# Hand-Foot-and-Mouth Disease-Associated Enterovirus and the Development of Multivalent HFMD Vaccines

**DOI:** 10.3390/ijms24010169

**Published:** 2022-12-22

**Authors:** Xinglong Zhang, Yifan Zhang, Heng Li, Longding Liu

**Affiliations:** Key Laboratory of Systemic Innovative Research on Virus Vaccine, Institute of Medical Biology, Chinese Academy of Medical Sciences and Peking Union Medical College, Kunming 650118, China

**Keywords:** HFMD, multivalent vaccines, systemic immunity

## Abstract

Hand-foot-and-mouth disease (HFMD) is an infectious disease of children caused by more than 20 types of enteroviruses, with most cases recovering spontaneously within approximately one week. Severe HFMD in individual children develops rapidly, leading to death, and is associated with other complications such as viral myocarditis and type I diabetes mellitus. The approval and marketing of three inactivated EV-A71 vaccines in China in 2016 have provided a powerful tool to curb the HFMD epidemic but are limited in cross-protecting against other HFMD-associated enteroviruses. This review focuses on the epidemiological analysis of HFMD-associated enteroviruses since the inactivated EV-A71 vaccine has been marketed, collates the progress in the development of multivalent enteroviruses vaccines in different technical routes reported in recent studies, and discusses issues that need to be investigated for safe and effective HFMD multivalent vaccines.

## 1. Introduction

Hand-foot-and-mouth disease (HFMD) is mainly caused by enteroviruses (EVs), which are single positive-stranded RNA viruses structured by a highly-structured 5′-untranslated region (5′ UTR), one open reading frame (ORF), and a 3′ UTR with a poly(a) tail [1,2]. The 5′ UTR comprises an RNA cloverleaf structure followed by an internal ribosomal entry site (IRES) [3,4]. The IRES is a highly-structured RNA region that directly recruits ribosomes for viral protein translation in a cap-independent manner [5]. The genome size of EVs is approximately 7.4 kb and encodes a single polyprotein of approximately 2100 amino acids, divided into three subregions, P1, P2, and P3, respectively. The P1 region encodes four structural proteins (VP4-VP2-VP3-VP1), and the P2 and P3 regions encode seven nonstructural proteins (P2-2A, 2B, 2C; P3-3A, 3B, 3C, 3D) [6]. Four structural proteins assemble to form the basic building blocks of the virion capsid named the protomer. Five protomers aggregate together to form a pentamer, and 12 pentamers plus the viral genome form an icosahedral virion approximately 30 nm in diameter [7,8].

HFMD mainly occurs in children under the age of five and is associated with sores in the mouth, anorexia, low-grade fever, and minor blisters or small ulcers on the hands, feet, and mouth [9,10]. Most children recover spontaneously in about a week, though sometimes the infection can cause complications such as viral myocarditis, pulmonary edema, aseptic meningoencephalitis, and cognitive impairment mediated by neuroinflammation [11]. Moreover, HFMD develops rapidly in individual children with severe diseases, leading to death [12]. On 3 December 2015, the China Food and Drug Administration (CFDA) approved the first EV-A71 vaccine to prevent HFMD. This feat represents considerable progress in preventing and controlling HFMD, the first in human history [8,13]. The EV-A71 vaccine has played an essential role in controlling the HFMD epidemic caused by EV-A71-related infection; however, it lacks the ability to provide cross-protection against other enteroviruses that cause HFMD, making the control of HFMD a significant challenge; thus, more broad-spectrum and efficient vaccines are needed [14]. This review focuses on the changes in the epidemiological spectrum of HFMD-associated enteroviruses post the EV-A71 inactivated vaccine using the research progress of multivalent vaccines for HFMD, and potential scientific issues regarding the inducing of systemic and balanced immune response in body, which is essential for development of new high-efficiency and broad-spectrum multivalent vaccines.

## 2. Changes in the Epidemiological Spectrum of HFMD-Associated Enteroviruses after the EV-A71 Inactivated Vaccine Was Approved for Marketing

Group A enteroviruses are the main enteroviruses that cause HFMD [15], among which EV-A71, CV-A16, CV-A10, and CV-A6 are the main virus-causing strains [16,17,18]. The enterovirus B-family can also cause some sporadic cases of HFMD [19,20], among which CV-B3 and CV-B5 infections that often cause more severe symptoms such as viral myocarditis and aseptic meningitis [21,22], so they are closely monitored.

HFMD has been included in Class C infectious disease management in China since 2 May 2008. According to the Report on Notifiable Infectious Diseases from the Chinese National Center for Disease Control and Prevention (www.chinacdc.cn, accessed on 10 October 2022), from 2009 to 2022 the annual morbidity and mortality of HFMD in China ranked among the top three in Class C infectious diseases (Figure 1a,b), excluding the lower morbidity numbers in 2020 and 2021. This may have been due to a series of COVID-19 pandemic prevention and control measures, such as home quarantine and school closures. In other years, the number of HFMD morbidity cases remained at approximately 2 million; it did not show a continuous decreasing trend, even though the first inactivated EV-A71 vaccine was approved at the end of 2015. However, according to relevant epidemiological findings, the number of cases of morbidity, severe illnesses, and deaths associated with EV-A71 have decreased significantly in China, with 62.8% of severe HFMD illnesses caused by EV-A71 from 2013 to 2015, while this percentage increased to 67.2% of severe HFMD illnesses caused by non-EV-A71 enteroviruses from 2017 to 2019 (Figure 1c) [23]. However, EV71 is still the most prevalent serotype in unvaccinated areas, such as Vietnam and Korea (Figure 1d) [24,25]. These observations indicate that the inactivated EV-A71 vaccine has played a key role in controlling EV-A71-associated HFMD.

The HFMD cases remains higher after EV-A71 inactivated vaccination, implying that an investigation on prevalence of non-EV-A71 enteroviruses is imperative. Examples include China and the Chengdu and Kunming areas. The prevalence spectrum of HFMD virus changed significantly after EV-A71 inactivated vaccination (Figure 2a–d) [26,27,28,29]; a noteworthy indication of the fact that the prevalence of CV-A6, CV-A10, and CV-A16 have significantly exceeded EV-A71.

## 3. Immunological Response to HFMD-Associated Enterovirus Infection

When a virus infects the body, a series of molecules and cells involved in the immune system are mobilized to fight the infection, while the virus uses its molecular features to influence the regular activity of sensitive cells—or immune cells—to obtain better living conditions and proliferation [30,31]. This interaction between the virus and the immune system involves a series of complex immune regulatory responses that often determine the final pathological outcome of viral infections [32,33,34]. Therefore, an in-depth study and understanding of these immune regulatory responses can aid in the development of appropriate viral vaccines and in their evaluation.

### 3.1. Native Immune Response Induced by a Viral Protein of HFMD-Associated Enterovirus

During viral infection, viral components such as viral proteins and nucleic acids act as pathogen-associated molecular patterns (PAMPs), which are recognized by pattern-recognition receptors (PRRs) to trigger antiviral innate immune responses [35,36,37]. For example, for the influenza virus several PRRs, such as endosomal TLRs (TLR3, TLR7/8, and TLR9) and cytosolic RLRs (RIG-I and MDA5), recognize viral nucleic acids to trigger downstream signaling pathways, resulting in the induction of the production of type I IFNs and inflammatory cytokines [38,39,40]. However, there are few reports regarding the native immune response induced by viral nucleic acids of HFMD-associated enteroviruses. This phenomenon may be due to two reasons. The first is that viral replication organelles (ROs) may provide a safety platform for HFMD-associated enterovirus RNA synthesis to escape the innate immune sensors that can detect viral RNA [41,42]. Another may be due to the characteristics of their RNA structure. Therefore, the native immune response induced by HFMD-associated enterovirus is mainly due to viral proteins [43,44,45]. Viral proteins consist of structural and nonstructural proteins [18]. It is more common for nonstructural proteins to be involved in the native immune response than it is for structural proteins to be involved (Table 1).

### 3.2. Adaptive Immune Response Induced by a Viral Protein of HFMD-Related Enterovirus

As previously mentioned, all HFMD-associated enteroviruses are structured by four different structural proteins (VP1–VP4). During assembly, the P1 polyprotein is cleaved into VP0, VP1, and VP3, and then VP0 is subsequently cleaved into VP2 and VP4. VP1–VP3 are exposed on the capsid surface while VP4 is located inside the capsid [66]. The VP1, VP2, and VP3 proteins mainly involve immune responses and host-receptor binding. VP1 contains the major neutralization epitopes and is often used for viral identification and evolutionary analyses [67]. Antigenic epitopes of VP2 and VP3 play an important auxiliary role in the process of antiviral infection [68]. The obvious differences in the amino acid sequences in these structural proteins indicate that distinct arrangements of loops among different serotype enteroviruses can create unique antigenic epitopes and may explain why the monovalent vaccine does not induce a cross-neutralization response against other HFMD-associated enteroviruses. Clearly, targeting individual EV-A71, CV-A16, CV-A6, and CV-A10 viral antigens does not provide enough cross-immunity; hence, a multivalent HFMD vaccine is needed to induce broad protection against each individual virus [69].

### 3.3. Immune Response to Dendritic Cells Infected with HFMD-Associated Enterovirus

Hu et al. used rhesus monkey CD1c^+^ DCs as a cell model to study the sensitivity of EV-A71 and CA16 to dendritic cells (DCs). They compared the differences in the levels of IFN-I production pathway-related molecules and Th-cell differentiation-related molecules induced after EV-A71 and CA16 infection. They found that both EV-A71 and CA16 could infect CD1c^+^ DCs. The difference was that the expression of IFN-I significantly decreased after EV-A71 infection of CD1c^+^ DCs, the viral load and viral titer increased, and the expression of Th2 and follicular helper T-cell (Tfh)-related transcription factors and cytokines increased significantly. In contrast, the expression of IFN-I increased significantly after CA16 infection of CD1c^+^ DCs, the viral load and viral titer decreased, and Th1 and regulatory T-cell (Treg)-related transcription factors and cell factor expression increased significantly. Additionally, the rhesus monkey infection experiment revealed that EV-A71 can induce the production of specific neutralizing antibodies after infecting infant rhesus monkeys, but CA16 cannot. This difference could be due to the main functions of Th2 and Tfh cells in CD4^+^ T-cell subsets assisting B cell proliferation, differentiation, antibody production, and participation in humoral immune responses, while the main functions of Th1 and Treg cells in CD4^+^ T-cell subsets are mediating cellular immunity and exerting immunosuppressive effects. Similarly, some researchers performed miRNA gene expression profiling and found that EV-A71 can use host miRNAs to regulate the host immune system and enhance cell survival. However, the knockdown of miRNAs resulted in significant upregulation of many antiviral-related signaling molecules, such as TLRs, RLRs, NOD-like receptors (NLRs), and IFN-I, in the host [70]. This finding suggests that EV-A71 can escape the body’s intrinsic immune-related signaling pathways during infection. In addition, Zhang et al. confirmed that EV-A71 and CA16 exhibited different effects on the expression of molecules involved in the interferon gene pathway when rhabdomyosarcoma (RD) cells were infected, and EV-A71 inhibited IFN-α/β-mediated JAK-STAT signaling and the expression of ISGs in these infected cells. However, CA16 infection had the opposite effect [71].

DCs, as key professional antigen-presenting cells (APCs), take up antigens via innate immune receptors (PRRs) and process them internally to create short peptide segments that are loaded onto the platforms of MHC class I and MHC class II proteins. These complex antigens, made up of MHC proteins plus the peptides they carry, are then recognized by T cells. Reduced MHC expression is a common mechanism through which viruses evade cytotoxic T cells to promote virus proliferation [72,73]. Yiwen Zhang et al. reported that open reading of frame 8 (ORF8) in SARS-CoV-2 can interact directly with MHC-Ι molecules and mediate their decreased expression [74,75]. In addition, Wonseok Kang et al. found that IFN-induced expression of MHC class I is attenuated in hepatitis C virus (HCV)-infected cells through the activation of PKR [76,77,78]. Whether enterovirus can affect MHC expression after the infection of DCs has not been reported.

## 4. The Immune Effect of the Experimental HFMD-Associated Enterovirus Multivalent Vaccine Reported in Recent Studies

To date, only the inactivated EV-A71 vaccine has been successfully approved for marketing as an HFMD-associated enterovirus vaccine; however, the inactivated EV-A71 vaccine lacks the ability to provide cross-protection against other HFMD-associated enteroviruses, and the epidemiological spectrum of HFMD-associated enteroviruses has shifted significantly since the approval and marketing of the inactivated EV-A71 vaccine. As the increasing number of pathogenic strains and the desire for achieving the smallest number of injections during vaccination have promoted the development of multivalent vaccines [7], all HFMD-associated enteroviruses have similar structural features, and no interaction effect has been reported during coinfection with these enteroviruses. Therefore, most HFMD vaccines currently studied have been developed in the form of multivalent vaccines but are only in the research phase of animal testing.

### 4.1. EV-A71/CV-A16 Bivalent Vaccine

The EV-A71/CV-A16 bivalent vaccine reported in recent studies is mainly in the form of an inactivated vaccine. Since CV-A16 and EV-71 have very similar viral capsid structures and biological characteristics, the receptors used for infection are also the same for both viruses. Of note, there has been no report regarding antibody-dependent enhancement occurring during infection with both viruses. In addition, the previously approved inactivated EV-A71 vaccine can induce highly effective immune protection in the body. Based on the points above, the EV-A71/CV-A16 bivalent inactivated vaccine has been considered to have a high potential for marketing. In 2017, Wang et al. immunized rhesus monkeys using an inactivated CV-A16 vaccine with the same preparation and vaccination method as that used for the inactivated EV-A71 vaccine. The results indicated that macaques immunized with the CV-A16 inactivated vaccine surprisingly did not exhibit protection against a viral challenge, which was likely due to the inability of the CA16 antigen to induce an effective natural immune response by intramuscular inoculation [79]. Clinical patients with CA16 infection generally have low neutralizing antibody titers and some have experienced repeated infections. After observing this unique phenomenon, Fan et al., 2019, immunized rhesus macaques with two doses of bivalent EV-A71-CA16 inactivated vaccine via the intradermal route. The immunization effect was examined longitudinally and included assessments of clinical symptoms, viral shedding, neutralizing antibodies, IFN-γ-specific ELISpots, and tissue viral load. Protection against EV-A71 and CA16 challenge was observed in all immunized macaques with no immunopathological effects [80,81].

### 4.2. Bivalent CV-A6/CV-A10 Vaccine

In 2017, Zhang et al. reported the protective efficacy and immunogenicity of a CVA6/CVA10 bivalent formaldehyde-inactivated vaccine. They found that subcutaneous delivery of the bivalent vaccine can induce antigen-specific systemic immune responses, particularly the induction of the production of polyfunctional T cells, which elicit active immunization to achieve a protection rate of >80% in infected neonatal mice. In addition, passive transfer of the antisera from immunized mice efficiently protected recipient mice against CVA6 and CVA10 challenges. In addition, because the immune response depends on the animal’s age and suckling mice are susceptible to infection with coxsackieviruses, such as CVA6 and CVA10, maternal antibodies are essential for assessing the vaccine’s effectiveness. It was found that maternal antibodies significantly reduced the degree of tissue lesions and viral load in neonatal mice [82].

### 4.3. EV-A71/CV-A6/CV-A10 Trivalent Vaccine

Elizabeth A et al. found that the combined application of viruses adaptation and use of IFN-deficient mice can enhance the age of EV-A71 susceptibility in 12-week-old mice [83]. With the same method, they produced mouse-adapted strains of the CVA16 and CVA6 viruses by sequentially passing the viruses in mice lacking interferon (IFN)α/β(A129) and α/β and γ(AG129) receptors. Finally, they used these models to evaluate the effect of a trivalent vaccine consisting of inactivated EV-A71, CVA16, and CVA6 in active and passive immunization studies. Total protection from a lethal challenge against EV-A71 and CVA16 was observed in the trivalent vaccinated groups. However, the mouse models used were interferon receptor-deficient mice, and their validity and feasibility for viral infection and vaccine evaluation, require further observance [84].

### 4.4. EV-A71/CV-A16/CV-A6/CV-A10 Tetravalent Vaccine

Since 2012, Huang’s laboratory has successively validated that EV-A71-VLP, CVA16-VLP, and CVA6-VLP generated by the baculovirus-insect cell expression system show sound protective effects and immunogenicity in their respective mouse models. In 2018, they obtained CVA10-VLP and then combined it with EV-A71-VLP, CVA16-VLP, and CVA6-VLP to generate a tetravalent VLP vaccine. Mouse immunization studies demonstrated that the tetravalent vaccine elicited antigen-specific and durable serum antibody responses. Importantly, passively transferred tetravalent vaccine-immunized sera provides valid protection against single or mixed infections with EV-A71, CVA16, CVA10, and CVA6 viruses in mice [85]. In contrast, the monovalent vaccines could only protect mice from the corresponding virus, but not from challenge by other serotypes. However, the immunogenicity of many vaccines exhibits differences in nonhuman primates and mice; for instance, EV-A71-VLP was found to induce the production of lower neutralizing antibody titers than inactivated EV-A71 in macaque monkeys but was more potent at inducing the production of neutralizing antibodies and showed better protection in mice [86]. Therefore, it is necessary to test the immunogenicity of this tetravalent vaccine in nonhuman primates.

### 4.5. CVB Hexavalent Inactivated Vaccine

The CVBs contain six serotypes, CVB1–6. CVBs are known to cause encephalitis and aseptic meningitis, myocarditis, chronic dilated cardiomyopathy (DCM), and pancreatitis. However, they can also cause HFMD. In 2020, V.M. Stone et al. reported a hexavalent CVB vaccine consisting of formalin-inactivated CVB1–6 viruses and tested its immunogenicity in mice and nonhuman primates. The results showed that this hexavalent CVB vaccine can elicit valid neutralizing antibody responses to the six serotypes and had an excellent safety profile in both animal models without an adjuvant. They also demonstrated that this vaccine provides immunity against acute CVB infections in mice, but the vaccine also needs to be assessed based on clinical trials to evaluate its immune protective effect [87,88].

## 5. Discussion

The development of multivalent vaccines is a systematic project that requires consideration of the physical compatibility, stability, potential immunological interference, and immune balance [89]. Although the successful development of EV-A71 inactivated vaccines has laid a good foundation for HFMD multivalent vaccines, there are still many issues that need to be resolved. For these problems, we offer the following solutions.

### 5.1. Enhancing the Immune Protection of the HFMD Multivalent Vaccine by Inducing an Effective Innate Immune Response

From 2015 to the present, the EV-71 inactivated vaccine has been the only licensed vaccine targeting HFMD, and the development of a vaccine for other serotypes remains challenging. For example, since immunized rhesus monkeys failed to provide effective protection in viral challenge trials, there is no CA16 vaccine available for clinical trials at this time. In addition, in 2018, Zhang et al. reported that CVA10-neutralizing and CVA6-neutralizing titers for the monovalent CVA10-VLP (GMT = 406) and the monovalent CVA6-VLP (GMT = 256) groups, respectively, were generally lower than the EV-A71-neutralizing and CVA16-neutralizing titers induced by the corresponding monovalent EV-A71-VLP (GMT = 2896) and monovalent CVA16-VLP (GMT = 5793) vaccines [85]. In 2019, Fan et al. conducted research focusing on innate immune responses elicited by inactivated EV-A71 and CA16 antigens administered intradermally or intramuscularly and found several interesting differences after comparison and analysis [80]. First, the expression levels of NF-κB pathway signaling molecules, which were capable of activating DCs, ILCs, and T cells, were higher in the intradermal group than in the intramuscular group. Second, the rates of antigen and ILC/DC colocalization in the intradermal groups were obviously higher than those in the intramuscular groups. Interestingly, these colocalization characteristics were associated with the time course of the experiment. ILC colocalization decreased over time, while DC colocalization increased over time. The third was that ILCs coordinated with DCs to activate T-cell proliferation more effectively (Figure 3). These results indicate that immunization through the intradermal route may help to elicit effective immunity especially against CA16. Therefore, they immunized mice with inactivated EV-A71 and CA16 antigens via the intradermal and intramuscular routes, respectively, and the results showed that immunization with EV-A71 and/or CA16 antigens via the intradermal route significantly increase neutralizing antibody titers and activate specific T-cell responses more than immunization via the intramuscular route. In addition, neonatal mice born to mothers immunized with the EV-A71 and CA16 antigens were 100% protected against wild-type EV-A71 or CA16 viral challenge. More importantly, the intradermal administration of a bivalent EV-A71-CA16 inactivated vaccine in rhesus macaques also induced specific immunity against EV-A71 and CA16 [81]. These results suggest the potential for developing this bivalent EV-A71-CA16 vaccine.

The above work provides two essential considerations for developing an HFMD multivalent vaccine in the future. The first is that intradermal immunization may enhance the immune effect of the HFMD multivalent vaccine. Therefore, the promotion and improvement of intradermal injection is an important point worth exploring. The second is that the coordination between DCs and ILCs contributes to successful adaptive immunity against vaccine antigens in the skin. Nevertheless, an understanding regarding the mechanism of ILC and DC coordination is not precise; thus, identifying the specific cell subtype and immune signaling molecule or pathway involved in this coordination is the next step.

### 5.2. Balancing Immunogenicity between Antigens to Reduce the Risk of Immune Interference

It is well known that the development of multivalent vaccines is not easy because of immune interference, during which one antigen in the multivalent vaccine is dominant over the others, resulting in imbalanced immune responses and insufficient protection against target pathogens. To date, no apparent antigenic interference has been reported in the development of HFMD multivalent vaccines, but some phenomena still need attention and further investigation. In 2016, Liu et al. found that an inactivated tetravalent EV-A71/CVA16/CVA10/CVA6 vaccine exhibited obvious differences in inducing the production of neutralizing antibodies against all four viruses in a mouse model; neutralizing antibody titers were 708 for EV-A71, 22 for CVA16, 16 for CVA10, and 100 for CVA6 [90]. The exact mechanisms underlying this result are still not precisely known; therefore, two experiments may help to further characterize the immunogenicity of antigens in HFMD multivalent vaccines. The first is a systematic comparison of immunogenicity generated by the multivalent vaccine and the corresponding monovalent vaccine to determine whether HFMD multivalent vaccines containing different serotypes of enteroviruses impact each other. The second would be to conduct a more detailed analysis of the immunogenicity of the HFMD multivalent vaccine in vivo and in vitro that explores a method for maintaining a relative balance between different antigens, such as by adjusting the dose rate of different antigens in the HFMD multivalent vaccine and using adjuvants to improve the immunogenicity of weaker antigens (Figure 4).

### 5.3. Inducing a Broad Spectrum of Protective Immune Response Coping Changes in the Epidemic Spectrum of HFMD-Associated Enterovirus

As mentioned previously, the epidemiological spectrum of HFMD-associated enteroviruses varies every several years. Therefore, the ideal HFMD multivalent vaccine elicits a broad-range of immune protective responses against HFMD-associated enterovirus. Multiepitope-based peptide vaccines carrying conserved CD4^+^, CD8^+^ T-cell, and B-cell epitopes among HFMD-associated enteroviruses may induce a broad spectrum of protective immune responses.

Neutralizing epitopes are a vital antigen component of an effective vaccine [91]; the neutralization epitope of EV71 have been screened and reported extensively due to their prevalence rate and severity in recent years. The structural proteins of EV-A71 and CV-A16 share approximately 80% sequence identity, and both viruses use the same cellular receptors [31]. Thus, most of the neutralizing epitopes of CV-A16 are similar with EV71.

Even though there is quite a difference in amino acid sequence among EV-A71, CV-A16, and CV-A6, they also have common neutralizing epitopes on the exact domains of VP1, VP2, and VP3 [92]. In addition, a bioinformatics-based prediction may be an efficient auxiliary tool to identify more neutralizing epitopes among HFMD-associated enteroviruses.

In 2015, Yueh-Liang Tsou et al. developed a recombinant adenovirus vaccine Ad-EVVLP with the EV-A71 P1 and 3CD genes inserted into the E1/E3-deleted adenoviral genome. Immunogenicity studies in mice have shown that antisera immunized with Ad-EVVLP can effectively neutralize the B4 and C2 genotypes of EV-A71, while immunization with Ad-EVVLP induces the production of 3C-specific CD4+ and CD8+/IFN-γ T cells, which could mediate protection against CVA16 challenge. Therefore, they continued to generate Ad-3CD expressing only the 3CD gene and immunized hSCARB2-Tg mice followed by EV-A71 or CVA16 challenge. The results demonstrated that Ad-3CD could fully protect immunized mice from EV-A71 and CVA16 challenges. Moreover, these results indicate that nonstructural proteins (such as 3 CD) of HFMD-associated enteroviruses may provide cross-immune protection [92]. Therefore, we aligned the amino acid sequences of all nonstructural proteins (2A, 2B, 2C, 3A, 3B, 3C, and 3D) of CV-A6, CV-A10, CV-A16, and EV-A71, and the homology was 97.33%, 97.96%, 97.24%, 97.57%, 98.12%, 97.78%, and 91.35%, respectively, indicating that these nonstructural proteins were conserved in the main HFMD-related enteroviruses. Additionally, these nonstructural proteins play essential roles in the viral replication cycle. Patients with severe EV-A71 infection usually have lower levels of cellular immune response compared to patients with mild disease, whereas neutralizing antibodies did not show significant differences in mild and severe cases. These studies suggest that cellular immunity may be critical for protection against enterovirus infection. According to the above comprehensive analysis, using conserved nonstructural proteins and the same neutralizing epitopes among HFMD-associated enteroviruses as antigens may induce a broad spectrum of protective immune responses (Figure 4).

## 6. Conclusions

HFMD is a public health threat with high morbidity and infectivity, especially in the Asia–Pacific region. The Report of Notifiable Infectious Diseases from the Chinese National Center for Disease Control and Prevention (www.chinacdc.cn, accessed on 10 October 2022) indicates that from 2009 to 2022, the annual morbidity and mortality of HFMD in China consistently ranked among the top three in Class C infectious diseases. The first inactivated enterovirus A71 (EV-A71) vaccine was approved and widely applied in China. From then henceforth, the EV-A71-related morbidity and severity decreased significantly in China, whereas EV-A71 is still the most prevalent serotype in unvaccinated areas, such as Vietnam and Korea. These results indicate that the inactivated EV-A71 vaccine is an effective tool for controlling EV-A71-related HFMD. However, HFMD-related viruses include many serotypes, and there has been a remarkable change in the epidemiological spectrum of HFMD-associated enteroviruses post the EV-A71 inactivated vaccine using the prevalence of CV-A6, CV-A10, and CV-A16 have significantly exceeded EV-A71 in Kunming, Yunnan Province. To effectively control HFMD, developing a multivalent vaccine that effectively targets HFMD-associated major virus (EV-A71, CV-A16, CV-A10, CV-A6) is the current trend. We analyzed and summarized three critical problems in the current HFMD multivalent vaccine studies to learn how to induce an effective innate immune response as well as how to induce a balanced immune response between antigens in order to reduce the risk of immune interference. In addition, our analysis included how to induce a broad spectrum of protective immune response coping changes in the epidemic spectrum of HFMD-associated enterovirus. In response, we provide the corresponding potential solutions as follows: Promotion and optimization of intradermal injections or in-depth study of molecules and signaling pathways involved in intradermal innate immune responses.; adjustment of dose rates of different antigens in HFMD multivalent vaccine and use of adjuvants to improve the immunogenicity of weaker antigens, and a combination of the conserved nonstructural proteins and the same neutralizing epitopes among HFMD-associated enteroviruses as antigens to design HFMD multivalent vaccine.

The development of multivalent vaccines is a systemic project involving multiple components. In addition to the several important issues mentioned above, there are some challenges that need to be overcome. Since different serotypes of HFMD-associated enteroviruses are not equally sensitive to the same animal, the first challenge is how to construct a suitable animal model to evaluate the safety and efficacy of the HFMD multivalent vaccine. Additionally, building a network for epidemiological investigation of HFMD-associated enteroviruses across regions or even countries remains a critical challenge. This will facilitate the selection of virus strains for HFMD multivalent vaccine research and the subsequent evaluation of vaccine efficacy.

## Figures and Tables

**Figure 1 ijms-24-00169-f001:**
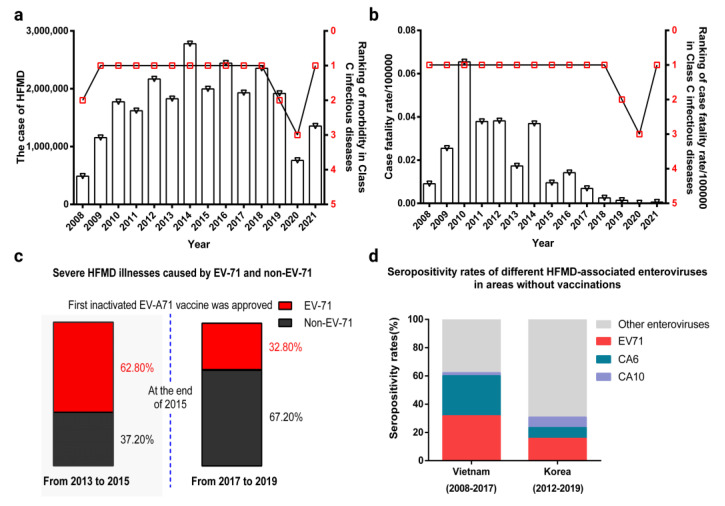
The epidemiological data of HFMD. (**a**) The number of HFMD cases in China from 2008 to 2021 and the ranking of its morbidity among C infectious diseases. The data in the figure are cited from www.chinacdc.cn (accessed on 10 October 2022). (**b**) The fatality rate of HFMD in China from 2008 to 2021 and its ranking among category C infectious diseases. The data in the figure are cited from www.chinacdc.cn (accessed on 10 October 2022). (**c**) The rate of severe HFMD illnesses before and after the first inactivated EV-A71 vaccine was approved in China. The data in the figure are cited from [23]. (**d**) Seropositivity rates of different HFMD-associated enteroviruses in Vietnam and Korea. The data in the figure are cited from [24,25].

**Figure 2 ijms-24-00169-f002:**
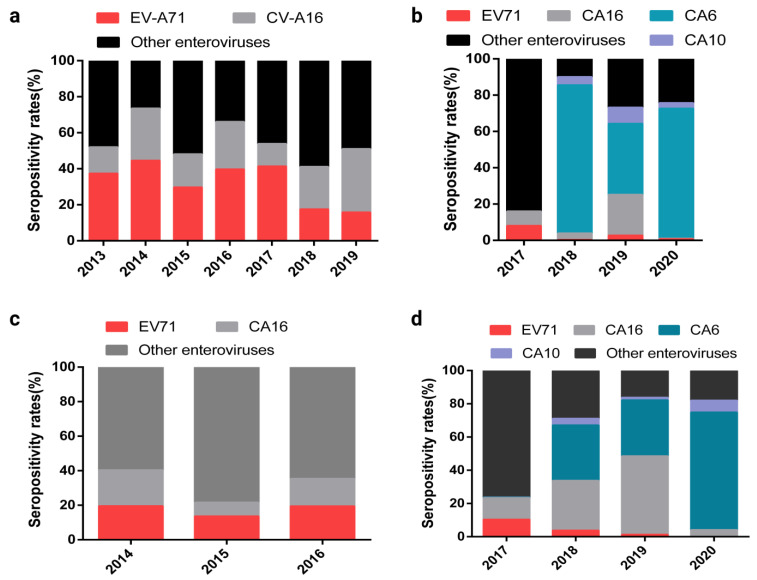
Seropositivity rates of different HFMD-associated enteroviruses in the whole of China along with the Chengdu and Kunming areas, 2013–2020. (**a**) Seropositivity rates of different HFMD-associated enteroviruses in the whole of China, 2013–2019. The data in the figure are cited from [26]. (**b**) Seropositivity rates of different HFMD-associated enteroviruses in Chengdu, Sichuan province, China. The data in the figure are cited from [27]. (**c**) Seropositivity rate of different HFMD-associated enteroviruses before EV-A71 inactivated vaccine vaccination in Kunming, Yunnan Province, China. The data in the figure are cited from [28]. (**d**) Seropositivity rate of different HFMD-associated enteroviruses after EV-A71 inactivated vaccine in Kunming, Yunnan Province, China. The data in the figure are cited from [29].

**Figure 3 ijms-24-00169-f003:**
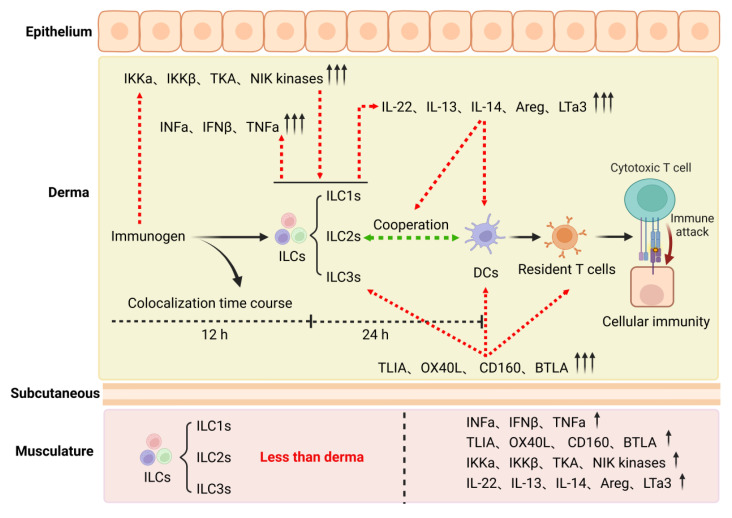
The potential mechanism that intradermal immunization is more effective than intramuscular immunization. Image created with BioRender (https://biorender.com/, accessed on 10 October 2022).

**Figure 4 ijms-24-00169-f004:**
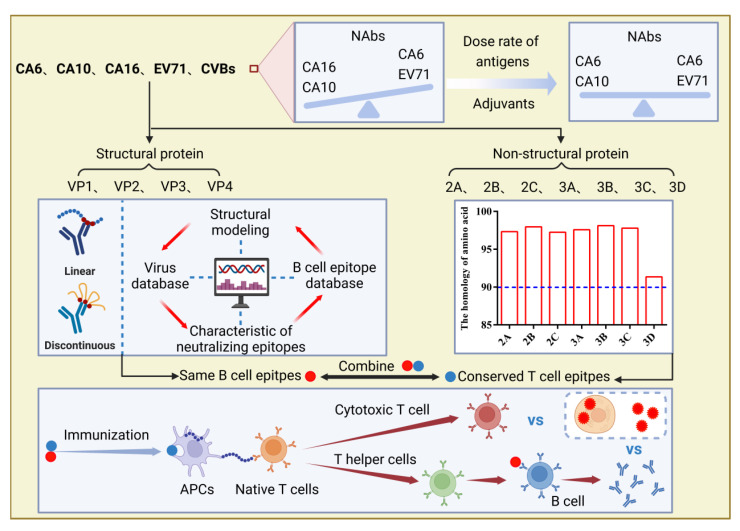
A series of potential methods to get more balanced immune responses and a broader spectrum of protective immune responses in development of multivalent HFMD vaccines. Image created with BioRender (https://biorender.com/, accessed on 10 October 2022).

**Table 1 ijms-24-00169-t001:** Native immune response induced by HFMD-associated enterovirus nonstructural proteins.

Protein		Functions	Innate Immune Response	Re
EV-71	2A	Cleaves NLRP3 protein Cleaves IFN-α/β receptor 1(IFNAR1) Cleaves the nuclear pore glycoprotein protein 62 (Nup62) Slices mitochondrial antiviral signaling (MAVS) protein Reduces serine phosphorylation of signal transducers and activators of transcription 1 (STAT1)	Inhibits inflammasome activation Blocks IFN-induced Jak/STAT signaling Disrupts host nuclear transport pathways and alters nuclear permeability Inactivates the antiviral innate immune response of RIG-I Attenuates IFN-γ signaling	[46] [47] [48] [49] [50]
	2C	Interacts with the IPT domain of RelA(p65) Inhibits IKKβ activation	Reduces the formation of the predominant form of NF-κB(heterodimer p65/p50) Blocks NF-κB activation	[51] [52]
	3A	Interacts with the human β3 subunit of Na^+^/K^+^-ATPase (ATP1B3) protein,	Enhances the production of type-I IFN	[53]
	3C	Cleaves the TAK1/TAB1/TAB2/TAB3 Complex Cleaves gasdermin D and NLRP3	Suppresses cytokine expression Inhibits cell pyroptosis and NLRP3 inflammasome activation	[54] [55]
	3D	Binds to NLRP3 Decreases STAT1 expression	Activates NLRP3 inflammasome Attenuates IFN-γ signaling	[56] [32]
CA16	2C	Blocks the fusion of autophagosomes with lysosomes and triggers autophagosome accumulation	Induces autophagy	[57]
	3C	Binds to MDA5 and inhibits its interaction with MAVS	Blocks MDA5-triggered type I IFN induction	[58]
CA6	2C	Blocks the fusion of autophagosomes with lysosomes and triggers autophagosome accumulation	Induces autophagy	[59]
	3C	Binds to MDA5 and inhibits its interaction with MAVS	Blocks MDA5-triggered type I IFN induction	[60]
	3D	Binds to RIPK3	Induces RIPK3-dependent necroptosis	[61]
CB3	2A	Cleaves host protease ATG4A	Impairs autophagy	[62,63]
	3A	Binds to GBF1	interference with Arf1-mediated COP-I recruitment	[64]
	3C	Cleaves MAVS and TRIF	inhibits both the type I IFN and apoptotic signaling	[65]

## Data Availability

Not applicable.

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
