# Peer review of "Hand-Foot-and-Mouth Disease-Associated Enterovirus and the Development of Multivalent HFMD Vaccines"

_ijms, 2022, doi:10.3390/ijms24010169_

Round 1

Reviewer 1 Report

The review is devoted to a serious topic related to the pathogenic importance of enteroviruses in the development of HFMD infection in children, which is often associated with severe complications and even death. The review shows the possibilities of creating an effective vaccine to prevent this infection.

The authors have covered various aspects of HFMD quite exhaustively. The work could be recommended for publication without major changes and corrections.

Author Response

Dear Reviewer,

Thank you very much for your review comments. In order to make this review more complete and the data more solid, we have added a paragraph with the general conclusions, in addition, we added the whole of China (2013-2019) and Chengdu region (2017-2020) HFMD-associated enterovirus seropositivity data to Figure 2. Revised portions are marked in red on the paper, please see the attachment.            

Reviewer 2 Report

It is an interesting study, but it requires some additions.
You did not describe the working method.
Please, rephrase the sentence "The plot was generated from data
from." (line 83). Also, this formula is repeated twice în description in figure 2.
Also, I recommend you to add a paragraph with the general conclusions.

Author Response

Dear Reviewer,

Thank you very much for your review comments and suggestions. Those comments are all valuable and very helpful for revising and improving our paper. We have studied the comments carefully and have made corrections which we hope meet with approval. Revised portions are marked in red on the paper. The main corrections in the paper are as flowing:

Point 1: You did not describe the working method.

Response 1: This review focuses on the epidemiological analysis of HFMD-associated enteroviruses since the inactivated EV-A71 vaccine has been marketed, collates the progress in the development of multivalent enteroviruses vaccines in different technical routes reported in recent studies, and discusses issues that need to be investigated for safe and effective HFMD multivalent vaccines.

Point 2: Rephrase the sentence "The plot was generated from data
from." (line 83). Also, this formula is repeated twice in description in figure 2.

Response 2: We have rephrased the sentence to "The data in the figure are cited from"

Point 3: Also, I recommend you to add a paragraph with the general conclusions.

Response 3: We have added a conclusion based on the full text.

In addition, in order to make this review more complete and the data more solid, we have added the whole of China (2013-2019) and Chengdu region (2017-2020) HFMD-associated enterovirus seropositivity data to Figure 2. Please see the attachment    
